# Brain Energy Metabolism in Ischemic Stroke: Effects of Smoking and Diabetes

**DOI:** 10.3390/ijms23158512

**Published:** 2022-07-31

**Authors:** Ali Ehsan Sifat, Saeideh Nozohouri, Sabrina Rahman Archie, Ekram Ahmed Chowdhury, Thomas J. Abbruscato

**Affiliations:** Department of Pharmaceutical Sciences, School of Pharmacy, Texas Tech University Health Sciences Center, Amarillo, TX 79106, USA; ali.ehsan.sifat@gmail.com (A.E.S.); saeideh.nozohouri@ttuhsc.edu (S.N.); sabrina.archie@ttuhsc.edu (S.R.A.); ekramahm@buffalo.edu (E.A.C.)

**Keywords:** brain, diabetes, energy metabolism, ischemic stroke, smoking

## Abstract

Proper regulation of energy metabolism in the brain is crucial for maintaining brain activity in physiological and different pathophysiological conditions. Ischemic stroke has a complex pathophysiology which includes perturbations in the brain energy metabolism processes which can contribute to worsening of brain injury and stroke outcome. Smoking and diabetes are common risk factors and comorbid conditions for ischemic stroke which have also been associated with disruptions in brain energy metabolism. Simultaneous presence of these conditions may further alter energy metabolism in the brain leading to a poor clinical prognosis after an ischemic stroke event. In this review, we discuss the possible effects of smoking and/or diabetes on brain glucose utilization and mitochondrial energy metabolism which, when present concurrently, may exacerbate energy metabolism in the ischemic brain. More research is needed to investigate brain glucose utilization and mitochondrial oxidative metabolism in ischemic stroke in the presence of smoking and/or diabetes, which would provide further insights on the pathophysiology of these comorbid conditions and facilitate the development of therapeutic interventions.

## 1. Introduction

The human brain is capable of planning and executing complex behaviors, making decisions, and processing emotional and social conditions, for which the brain requires large amounts of energy [1]. Glucose plays a key role as a main energy substrate for both an adult brain and a developing brain. Oxidative metabolism supplies most of the energy required for maintaining brain activities [2]. Enhanced neuronal activity triggers increased energy consumption which may cause compensatory metabolic and vasculature changes to help maintain enhanced neuronal function [3]. Therefore, normal brain function needs tightly controlled energy metabolism both temporally and spatially from a regional level down to the level of a single synapse [4]. The neurovascular unit (NVU) is a recently developed concept in neuroscience which depicts the complex structural and functional relationship between brain and cerebral blood vessels [5]. A NVU consists of neurons, glial cells (astrocytes, microglia, and oligodendrocytes), and vascular cells (pericytes, endothelial cells, and vascular smooth muscle cells) [6,7]. Although neurons and astrocytes are key players in brain energy metabolism, the vascular cells of the NVU also play a crucial role in regulating brain energy metabolism. Glycolysis and oxidative phosphorylation are the preferred energy metabolism pathways for neurons and astrocytes, respectively. It has been observed in different studies that brain metabolic alteration is associated with the progression of different neurodegenerative disorders such as Alzheimer’s disease, amyotrophic lateral sclerosis, Parkinson’s disease, and Huntington’s diseases [8]. Moreover, a reduced level of glucose oxidation in post-ischemic brain tissue has been reported [9]. Another study demonstrated that energy-dependent neuronal processes comprise the delicate vulnerability of the brain to ischemia [10]. Therefore, it is evident that a disruption in brain energy metabolism may contribute to cerebrovascular and neurodegenerative dysfunctions including ischemic stroke.

Stroke is considered to be one of the leading causes of adult disability in developed countries and is the fifth leading cause of mortality in the Unites States (USA). In the USA, more than 795,000 people suffer from stroke each year and one person dies from stroke every four minutes [11]. Mainly, there are three types of strokes: ischemic, hemorrhagic, and transient ischemic attack, among which 87% strokes are ischemic [12]. Ischemic stroke can be caused by a major cerebral artery occlusion due to an embolus or clot which leads to temporary or permanent blood flow obstruction to the brain [13]. Impaired energy metabolism is a key pathological hallmark of ischemic stroke [14,15]. A reduction in glucose and oxygen supply results in severe loss of ATP production in an ischemic brain. Cerebral ischemia disrupts mitochondrial oxidative metabolism and enhances mitochondria-mediated oxidative stress. Neurons are more sensitive to ischemia, and they are depleted of ATPs more quickly than astrocytes [16]. Glial cells (astrocytes and microglia) are rapidly activated after ischemic stroke and they play an important role to regulate neuroinflammation [17]. 

There are several modifiable and non-modifiable risk factors associated with ischemic stroke. Among the modifiable risk factors, smoking and type 2 diabetes are the most common comorbid conditions for increased risk and poor outcome of stroke [13]. Tobacco smoke is a serious public health concern which is the leading cause of preventable disease and death in the USA [18]. Tobacco smoking can worsen ischemic stroke prognosis by increasing the permeability of the blood-brain barrier (BBB) and disrupting ion transporter function, thereby, enhancing edema formation in the brain [19,20]. Another risk factor underlying the pathogenesis of a stroke is diabetes which is the seventh leading cause of death in the USA. It has been demonstrated that diabetes is related to enhanced stroke-induced mortality rate [21]. Hyperglycemia can also worsen ischemic brain injury, edema formation, and post-ischemic seizures by enhancing oxidative stress and mitochondrial dysfunctions [22,23]. 

Several studies have shown that smoking/nicotine as well as diabetes may play pivotal roles in brain energy metabolism. Acute nicotine treatment has been shown to alter brain glucose utilization in most of the brain regions [24,25,26,27]. Additionally, nicotine may dysregulate mitochondrial function by generating reactive oxygen species (ROS) and increasing oxidative stress, which could be associated with the pathogenesis of cerebrovascular diseases including ischemic stroke [28]. In addition, the role of diabetes in brain energy metabolism has been investigated in several studies and it has been found that diabetes may decrease brain glucose utilization and cause neurological damage by altering BBB function, neurotransmitter metabolism, cerebral blood flow (CBF), and microvascular function [29,30]. Diabetes may downregulate glucose transporter 1 (GLUT1) expression and function at the BBB and decrease astroglial glucose metabolism, as observed in preclinical diabetic models [31,32]. Oxidative stress [33] and mitochondrial dysfunctions [23] are key factors contributing to hyperglycemia-induced enhanced ischemic brain injury. Thus, smoking and diabetes may both function as comorbid conditions and can simultaneously cause and worsen ischemic stroke outcomes by contributing to altered glucose utilization and mitochondrial dysfunctions. Therefore, the aim of this review article is to confer the effects of smoking and diabetes on brain energy metabolism and how the coexistence of these conditions could exacerbate ischemic stroke outcome by altering brain energy metabolism while pointing out the limitations of current knowledge. 

## 2. Brain Energy Metabolism

Although the human brain accounts for ~2% of the total body weight, it has very high energy requirements, accounting for at least 20% of the body’s energy consumption [34]. Tight coupling exists between the demand and supply of energy with changes in CBF and glucose utilization correlating with neuronal activity [35]. Therefore, an understanding of brain energy metabolism is crucial in understanding the physiology and pathology associated with the NVU. Classically, brain energy metabolism has been correlated with the supply of oxygen and glucose to the brain through CBF. Glucose has been considered to be the primary source of energy utilized by both neurons and astrocytes. The oxidation of glucose generates ATP which is needed for energy-dependent reactions. Lactate is a supplementary energy substrate for neurons [36]. Numerous studies have demonstrated that under particular conditions, such as fasting [37], uncontrolled diabetes [38], or maternal milk diet in newborns [39], ketone bodies are also able to meet the energetic requirements of the brain. 

### 2.1. Glucose Metabolic Routes 

Glucose typically enters neurons through GLUTs, and then is converted to glucose-6-phosphate by phosphorylation [40]. Then, glucose-6-phosphate can be processed through either of three different metabolic routes: glycolysis, pentose phosphate pathway (PPP) or, stored as glycogen. Glucose-6-phosphate metabolized through glycolysis gives rise to two molecules of pyruvate. Under normal conditions and in the presence of oxygen, pyruvate can enter mitochondria, where it is utilized in the tricarboxylic acid cycle (TCA) and oxidative phosphorylation to generate ATP. Under different conditions such as hypoxia or specific metabolic requirement of cells, pyruvate can also be reduced to lactate [41]. This lactate is transported to the extracellular space by monocarboxylate transporters (MCTs). The complete oxidation of glucose produces substantial amounts of energy in the form of 30–36 ATPs in the mitochondria as compared with glycolysis which produces only two ATPs. The PPP and glycolysis pathway are also both linked to the production of reducing equivalents in the form of NADPH [42]. NADPH is a critical defense component against oxidative stress through the metabolism of the tripeptide glutathione (GSH) [43]. GSH acts as an electron donor in several reactions, including the detoxification of ROS [44]. 

### 2.2. The Important Role of Astrocytes in Brain Energy Metabolism

The human brain consists of up to 10-fold higher number of glial cells as compared with neurons [45]. Astrocytes are the most prevalent glial cells which comprise almost 50% of the total human brain volume [46]. Morphologically, astrocytes are uniquely positioned to sense and respond to changes in neuronal activity allowing them to perform numerous essential functions [47,48]. These cells play a crucial role in maintaining brain ionic equilibrium, glutamate homeostasis, as well as the maintenance of ROS (in GSH recycling) and osmotic regulation [46]. Neurons rely on astrocytes for the supply of precursors of the Krebs cycle intermediates or their derivatives, as the enzyme pyruvate carboxylase is only present in astrocytes but not in neurons [49]. The energy demands of astrocytes comprise only about 10–15% of the total brain energy requirements. Approximately 85% of the glucose taken up in the brain is used in energy expenditure in neurons [50]. 

Astrocytes are the only cells in the brain capable of storing glycogen. The activity of pyruvate dehydrogenase (PDH), on the one hand, is high in neurons and low in astrocytes, resulting in limited capacity of glycolysis in neurons from the TCA cycle and the oxidative phosphorylation route being more active in those cells. Astrocytes, on the other hand, have more active glycolysis process that can be upregulated with limited pyruvate processing through the TCA cycle route. The high glycolytic rate of astrocytes suggests a preference for the production and release of lactate. Studies have demonstrated the neuroprotective role of lactate experimentally [51,52] which prevented the death of vulnerable neurons and slowed disease progression. 

### 2.3. Astrocyte–Neuron Lactate Shuttle 

Magistretti and Pellerin [53] proposed that astrocytes increase their rate of glucose uptake, glycolysis, and release of lactate into the extracellular space in response to intensified neuronal activity. The high neuronal activity releases glutamate into the extracellular space through the neuronal glutamate transporter, excitatory amino acid transporter 3 (EAAT3). This enhanced activity at the glutamatergic synapses is sensed by astrocytes who uptake the glutamate via glutamate transporters EAAT1 and EAAT2. The transport of glutamate is driven by a sodium gradient, where three Na^+^ ions are co-transported with one glutamate, resulting in a significant increase in intracellular Na^+^ concentrations in astrocytes. Glutamate in astrocytes is converted to glutamine, which is then released back into the extracellular space and taken up by neurons. Glutamine is converted back to glutamate in neurons, thereby, replenishing the pool of glutamate and completing the glutamate–glutamine cycle [35,54]. Glutamate uptake by astrocytes stimulates the uptake of glucose at a 1:1 stoichiometric relationship. The increase in Na^+^ concentrations in astrocytes through the glutamate-glutamine cycle activates the Na^+^-K^+^-ATPase, triggers glycolysis, and leads to the production and release of lactate into the extracellular space. This lactate can be utilized as an energy source by neurons for the generation of ATP through the TCA cycle and oxidative phosphorylation [55,56]. This interplay in glucose metabolism between neurons and astrocytes is illustrated in Figure 1. Angamo et al. [57] demonstrated that the astrocyte-neuron lactate shuttle (ANLS) regulates ion homeostasis and synaptic signaling in the presence of ample glucose. PDH is the rate-limiting enzyme that catalyzes lactate oxidation. The inactive form of the enzyme is present in greater proportions in astrocytes as compared with neurons [58]. Astrocytes can, therefore, be viewed as “lactate sources” producing and maintaining the extracellular lactate pool, while neurons can be considered to be “lactate sinks” that consume lactate in an oxidative manner to fulfill energy demands. 

The ANLS hypothesis has been thoroughly scrutinized over the years, where some studies criticized the concept [50,59]. Even though there is an ongoing debate, overwhelming experimental evidence supports the ANLS hypothesis [60,61,62]. It is now widely accepted that lactate is a crucial component in the brain energy metabolism process. 

## 3. Brain Energy Metabolism in Cerebral Ischemia

Disruption in brain energy metabolism is a pathological hallmark of ischemic stroke [10]. A wealth of understanding has been gained related to changes after cerebral ischemia utilizing animal models of stroke based on the effects of pharmacological interventions/genetic modifications that drive brain injury/recovery. Various durations of middle cerebral artery occlusion (MCAO) in mice or rats have been used commonly in these investigations. The duration of the occlusion plays a critical role in determining the extent of injury and involvement of complex pathophysiological events. Due to the limited collateral perfusion in the cerebral arteries, ischemia is instantly developed in the brain tissue surrounding the occluded vessel. Blood flow is reduced by more than 80% in the core ischemic region [63]. GLUTs are upregulated at the brain endothelial cells and brain parenchyma [64,65] following ischemic stroke to meet the enhanced energy demand. Nevertheless, the disrupted supply of glucose and oxygen supply results in less ATP production. Ionic gradients across the plasma membrane also become depleted leading to a large potassium efflux out of the cells and calcium influx into the cells [66,67]. 

In brain tissue, the core of an injury is surrounded by penumbral tissue that has the capacity for full recovery. This metabolically unstable circumferential zone has 20–40% blood flow reduction and has normal K^+^ homeostasis. Neurons in the salvageable penumbra are hyperpolarized and electrically silent; however, their function could be recovered if the regional CBF is restored in time [63,68]. Therefore, if penumbra is not salvaged, it will progressively be recruited to the infarct core. In fact, the possibility of recovery or irreversible damage is determined by both the amount of residual blood flow and the duration of blood flow interruption within a region. The glucose and ATP content in the ischemic core falls markedly within the first 5 min of occlusion. ATP concentrations in the core region remain to be 15–30% as compared with non-ischemic tissue for at least the first 2 h of focal ischemia [69]. Phosphocreatine in the brain acts as a short-term energy reserve, allowing ATP regeneration from ADP in a reaction catalyzed by creatine kinase. Due to limitations in oxygen availability, glucose reaching the core tissue is metabolized via glycolysis to lactate resulting in a 10-fold or even higher levels of lactate accumulation [69,70]. The alterations in the penumbral tissue are less severe. Two hours after ischemic insult, phosphocreatine is reduced by 30% and ATP is reduced by 50% as compared with non-ischemic regions. Some of the produced ADP is metabolized further to generate AMP and ATP. This reaction is catalyzed by adenylate kinase and usually helps maintaining ATP level and meeting short-term energy requirements in the brain. AMP is also metabolized to inosine and hypoxanthine in ischemic brain tissue [71]. 

### 3.1. Role of Mitochondria in Ischemic Brain Energy Metabolism

Mitochondria play a crucial role in the alterations of brain energy metabolism and oxidative stress in ischemic stroke [72,73,74]. Glucose utilization in the penumbral area remains unaltered or increases in the initial 2 h of ischemic stroke. Increased glucose extraction from blood maintains glycolytic activity and increases the lactate concentration [75]. This suggests that oxidative metabolism of glucose is significantly impaired in penumbral brain regions, but glycolysis is mostly preserved. Reserve capacities cannot keep up with the increased energy demand of the penumbral tissue due to the reduction in oxidative metabolism [76]. Depolarized tissue proportionally loses ATP and phosphocreatine. Interestingly, even though mitochondrial respiratory function is significantly restored in the core and penumbral tissue within an hour after reperfusion, it declines at later time points. This secondary impairment in mitochondrial respiration seemingly develops earlier than the alterations in energy-related metabolites, when determined in the same ischemic model. It indicates that the delayed changes in mitochondrial function are an early step in the development of irreversible cell dysfunction and possibly a contributor to this process [72,73]. A lower membrane potential in mitochondria, isolated from core tissue, is observed when incubated under basal or ADP-stimulated conditions. This further indicates a decreased ATP generation capacity of mitochondria. Electron micrographs from brain tissue at 2 h of reperfusion after a 3 h duration of focal ischemia have shown significant structural defects in neuronal mitochondria, which was consistent with the functional impairment observed under similar conditions in other studies [77,78]. Neuronal A-kinase anchor protein 121 (AKAP121) has been observed to be degraded during focal brain ischemia and has been hypothesized to be partially responsible for the mitochondrial changes. This protein causes functional alterations in mitochondria in response to intracellular signaling. Degradation of AKAP121 hinders oxidative metabolism and decreases mitochondrial membrane potential [79]. 

Oxidative stress is a crucial factor in the development of both apoptosis, which is a programmed cell death pathway and necrosis, the unprogrammed cell death pathway which is predominant in ischemic core particularly after reperfusion [74]. In ischemic stroke, the mitochondria-mediated cell death pathway is considered to be an intrinsic apoptotic pathway rather than a receptor-mediated extrinsic apoptotic pathway. In the intrinsic pathway, different internal signals created by hypoxia, DNA damage, or oxidative stress are sensed by cells which results in programmed cell death [74]. Isolated mitochondria from the penumbra regions of rats during early recirculation showed increased calcium content and free radical production in vitro. These alterations were both significantly decreased in mitochondria from rats treated with dinitrophenol (oxidative phosphorylation uncoupler), suggesting a mitochondrial role in the protective effects [80]. GSH is a major water-soluble antioxidant localized in both the cytosol and the mitochondria of cells. Mitochondrial GSH levels reduce sharply during ischemia and are believed to be sufficient to induce infarct formation which persists following reperfusion [81]. Losses of mitochondrial GSH in vitro increase the susceptibility of astrocytes to oxidative stress [82,83]. It is important to note that the cell-type specific effects of ischemic stroke on mitochondrial dysfunctions in the NVU have not been fully elucidated and need further investigation. Mitochondria have been identified as an important therapeutic target in ischemic stroke treatment (Table 1) and several studies have suggested they play a key role in oxidative stress, inflammation, energy production, autophagy, mitophagy, lipid production, and overall stroke outcome.

### 3.2. Sources of Oxidative Stress in Cerebral Ischemia

Oxidative stress is a key player in the pathobiology of ischemic stroke [74], as described earlier. Hence, it is important to discuss the sources of oxidative stress in ischemic stroke. It has been observed that ROS play a critical role in cell homeostasis, blood flow, and the physiology of cerebral vasculature [115]. ROS are regulated in the body by cellular antioxidant systems including GSH, superoxide dismutase (SOD), glutathione reductase, glutathione peroxidase (GPx), antioxidant response element, catalase, and nuclear factor erythroid 2-related factor 2 coupled with NVU components containing neurons, astrocytes, pericytes, endothelial cells, and microglia [116,117]. ROS can affect microcirculation by changing blood flow resistance and affecting vascular pathophysiology in the brain [118]. Additionally, it has been observed that excessive ROS can disrupt BBB integrity and alter BBB permeability [119]. Microglia and astrocytes can produce an elevated level of ROS by the NADPH oxidase (NOX) pathway altering the expression of tight junction proteins (claudin-5, occludin, and ZO-1) of BBB. Pericytes play a significant role in maintaining BBB integrity and are susceptible to oxidative stress as well [120,121]. Activated microglia can produce ROS which can cause apoptosis of pericytes [122]. It has been demonstrated that ROS are also synthesized by immune cells such as polymorphonuclear neutrophils (PMNs) causing oxidation of important proteins required for cellular signaling such as tyrosine phosphatase which may lead to endothelial dysfunction. Previously, Kontos et al. reported the generation of superoxide during reperfusion. Meninges, vascular smooth muscle cells, and endothelial cells were identified as superoxide generation sites [123]. Neutrophils and macrophages are major components of the innate immunity systems which play a crucial role in inflammatory responses to infections and tissue injury by getting activated and transported.

Excessive amount of ROS can be generated after stroke, evidenced by the protective effects of ROS scavengers [124]. Oxidation of serum albumin has been found to be increased in stroke patients which may result in oxidation of amino acid residues by ROS [125]. A steady increase in ROS levels after MCAO model in in vivo study has also been observed after occlusion [126]. The second phase of ischemia/reperfusion (I/R) injury may be induced by reperfusion where ROS are generated [126]. One of the important sources of ROS generation is mitochondrial electron transport chain (ETC) [127]. During ischemia, the ubiquinone-cytochrome b region of the ETC is considered to be the key source for ROS generation [128]. A recent study identified succinate accumulated during ischemia as a potential mitochondrial metabolite that drove extensive ROS production [129]. Another crucial source of ROS generation during ischemic stroke is NOX enzymes. Elevated expression level of NOX has been found to be responsible for increasing MMP-9 upregulation after stroke, promoting BBB damage [130]. It has been observed that after ischemic stroke, NOX2 and NOX4 expression levels were upregulated [131]. Additionally, xanthine oxidase (XO) is considered to be another source of ROS generation during ischemic stroke. XO produces hydrogen peroxide which contributes to post-stroke brain edema after I/R [132]. In addition, XO mediated an increased level of superoxide anion radicals that could be observed in blood after forebrain I/R in experimental animal model [133]. Other intracellular enzymes involved in ROS production are cyclooxygenases (COX), lipoxygenases (LOX), and cytochrome P450 enzymes. Free arachidonic acid from cell membrane phospholipids is metabolized by the abovementioned enzymes and can contribute to generating superoxide during I/R [134]. Cerebral I/R injury has been found to be related to acute inflammation where both neutrophil and macrophages play a significant role. Rapid recruitment of neutrophils to ischemic areas and interaction of platelets with activated neutrophils may increase occlusion of vessels [135]. Reperfusion may increase oxygen in the tissue generating more ROS resulting in brain injury [136]. In addition, macrophages are activated in brain within a very short period of stroke onset and generate a range of proinflammatory mediators including TNFα and IL-1β which may worsen brain damage [137]. Figure 2 depicts the oxidative stress pathways and how they are perturbed in the setting of ischemic stroke. 

### 3.3. Metabolic Flexibility of Microglia: Potential Role in Ischemic Stroke

In addition to neurons, other cells of the NVU can regulate energy metabolism in the ischemic brain. Microglia are resident immune cells of the CNS, which play a crucial role in different pathological conditions [138,139]. A recent study by Bernier et al. demonstrated that microglia could alter their metabolic profile in glucose-deprived conditions to retain their immune surveillance functions for a long time [138]. The authors demonstrated that microglial immune surveillance remained functional in an in vivo model of hypoglycemia and an in situ model of aglycemia. They also showed that microglia could switch their preferred metabolic process from glycolysis in normal physiological conditions to glutaminolysis in aglycemic conditions. This effect could be critical for neuroprotection and vascular restoration in metabolically stressful conditions. Especially, the aglycemic component of ischemic stroke could be influenced by this metabolic flexibility of microglia. It would be interesting to see the effects of both oxygen and glucose deprivation on microglial immune functions and metabolic profile in the ischemic brain, which opens an exciting research area. 

## 4. Smoking and Diabetes as Comorbid Conditions for Ischemic Stroke

Comorbid conditions affecting energy metabolism in the ischemic brain need to be explored, as they may exacerbate the outcome of an ischemic event. Smoking and diabetes are considered to be critical risk factors and comorbidities for ischemic stroke. These comorbid conditions are associated with higher possibility of ischemic stroke occurrence and deteriorating brain damage, therapeutic consequence, and post-stroke recovery [13]. Smoking has been reported to be a risk factor for stroke in different studies involving diverse ethnicities and populations. It has been found that active smokers have two to four-fold enhanced stroke risk as compared with non-smokers or ex-smokers who had quit smoking more than 10 years ago [140]. Another study reported six-fold increased risk of stroke in smokers as compared with non-smokers who had never been exposed to second-hand smoke as well [141]. Moreover, a robust dose-dependent relationship has been found between smoking and cerebral ischemic stroke risk in young females which suggested a correlation between sex difference and stroke risk in the smokers [141]. Our laboratory has shown that nicotine exposure can worsen ischemic stroke outcome by increasing brain edema and infarct volume [20], and brain to blood potassium transport [19] in preclinical studies. Further, exposure to tobacco smoke and electronic cigarettes were shown to enhance brain injury and neurological outcome in an in vivo model of ischemic stroke [142].

There are three types of diabetes, namely type 1, type 2, and gestational diabetes. Type 2 diabetes is the most common type of diabetes and around 90–95% of diabetic patients suffer from type 2 diabetes [143]. Studies have found an association between diabetes and stroke. A total of 27% increase in the number of stroke patients with diabetes as comorbidity was reported in a study conducted from 1996 to 2006 [144]. The risk of ischemic stroke was around twice in diabetic patients as compared with non-diabetic patients [145]. A recent clinical study concluded that baseline hyperglycemia was associated with worsened clinical prognosis in acute ischemic stroke patients treated with intravenous thrombolysis [146], which was consistent with other studies [147]. Experimental studies have also reported that hyperglycemic conditions caused enhanced brain injury after ischemic stroke by decreasing the expression of the neuroprotective protein, alpha-synuclein [148], and worsening of microvascular thromboinflammation [149]. Denorme et al. also showed that hyperglycemia enhanced brain infarct size and BBB permeability, and worsened neurological outcomes and CBF [150]. Additionally, the effects of smoking and diabetes on NVU transporters have been reported in different studies, demonstrating the enhanced risk of stroke occurrence and worsened outcomes [13]. Alteration of brain energy metabolism could be an important mechanism of smoking- [151] and diabetes-induced [152] worsening of ischemic stroke prognosis. Below, we discuss how smoking and diabetes can alter brain energy metabolism, which may contribute to the pathobiology of ischemic stroke. 

## 5. Effects of Nicotine and Smoking on Brain Energy Metabolism

### 5.1. Glucose Utilization

The effects of nicotine and smoking on brain glucose utilization have not been fully elucidated yet, as conflicting reports exist on this matter. A dose-dependent increase of local cerebral glucose utilization (LCGU) in nine structures of the rat brain was observed in one study following administration of nicotine [153]. In this study, nicotine was infused in three dosages, i.e., 0.5, 1.58, and 5 μg/kg/min, which resulted in 10, 39, and 114 ng/mL plasma nicotine, respectively. Behavioral effects of nicotine on the central nervous system (CNS) were also reported, correlating the increase of LCGU in different parts of the limbic system [153]. Acute nicotine treatment (0.3 mg/kg, s.c.) increased LCGU [24]. Marenco et al. reported similar results in partially immobilized rats but found reduced global brain glucose utilization in freely moving rats with 0.4 mg/kg, s.c. nicotine treatment [25]. Restraining-induced stress could account for the increase in LCGU in some preclinical studies. A randomized, double blind, placebo-controlled study demonstrated that nasal administration of nicotine (1–2 mg) after overnight abstinence increased regional cerebral metabolism of glucose (rCMRglu) in the thalamus and visual cortex of humans. The thalamus was activated by nicotine, which could be due to the presence of nicotinic acetylcholine receptors (nAChRs) in high density [154]. In contrast, Stapleton et al. showed that acute nicotine administration (1.5 mg, i.v.) in humans decreased brain glucose utilization in most of the brain area [26]. A clinical study also showed that smoking could reduce CBF and CMRO_2_ after abstinence in chronic smokers (15 ± 5 cigarettes per day for 10 ± 5 years, each cigarette delivering 1 mg nicotine) which was acutely restored after smoking was resumed (2 ± 1 cigarettes) [155]. Later, Wang et al. showed a decrease in brain glucose oxidation with acute exposure of nicotine (0.7 mg/kg, s.c.) [27]. Nicotine contents for currently marketed tobacco products are 1.1–1.8 mg in tobacco cigarette and 0.5–15.4 mg/15 puffs in electronic cigarettes [156]. Plasma nicotine level after smoking a tobacco cigarette is 15 ng/mL [157]. The doses of nicotine in the abovementioned clinical studies correspond well with marketed cigarettes, while those of the preclinical studies may vary due to the physiological differences between humans and rodents. Different studies have reported that nicotine uptake resulted in altered glucose metabolism resulting in an increased level of blood sugar [158,159,160,161]. The cell type-specific effects of nicotine on normative brain energy metabolism are not known and need to be investigated. Further, the effects of nicotine on the vascular (endothelium, matrix, smooth muscle, astrocyte) and extravascular (astrocyte, neuron) relationships, in terms of energy metabolism need further research.

Few studies have investigated the effects of smoking on glucose utilization in the ischemic brain. Previous studies from our laboratory have shown that chronic nicotine exposure (4.5 mg/kg/day for 14 days) significantly reduced glucose transport across the BBB along with decreased expression of GLUT1 transporter in ischemic brain endothelial cells [151]. In addition, nicotine exposure (5 µM) decreased glucose uptake in neurons in ischemia-like condition [162,163]. In vivo exposure to nicotine-containing electronic cigarettes (2.4% nicotine) could also decrease brain glucose utilization and GLUTs’ expression (GLUT1 and GLUT3) in normoxic and ischemic condition [163]. A recent study showed that 16–21 days of smoking-derived nicotine treatment enhanced brain infarct volume after tMCAO in both adolescent and in adult female rats, which was linked with alterations in the glycolytic pathway [164]. These results suggest that nicotine, smoking, or vaping may lead to significant glucose deprivation in the ischemic brain, which could result in worsened outcome of an ischemic stroke event (depicted in Figure 3). It is noteworthy that some researchers have shown beneficial effects of activation of α7 nAChR subtype in ischemic stroke, which was mostly mediated by anti-inflammatory actions [165,166]. 

### 5.2. Mitochondrial Function and Oxidative Stress

The effects of nicotine and smoking on mitochondrial activity have been investigated in different studies including various experimental systems such as intact cell, isolated mitochondria, and animal model [167]. Cormier et al. reported that, nicotine inhibited oxygen utilization in isolated rat brain mitochondria [168]. Nicotine binds only to complex I of the mitochondrial respiratory machinery and prevents the flow of electron from NADH to complex I [168]. Another study demonstrated decreased levels of mitochondrial complex I, II, and III enzymatic activities after 7 days of nicotine treatment [169]. Wang et al. showed that chronic nicotine treatment affected the expression of a number of genes which encoded for mitochondrial respiratory complex subunits in different brain regions [170]. Another study showed that 16 days of nicotine treatment decreased mitochondrial energy metabolism in rat hippocampus which was linked with complex IV inhibition [171], while a recent study also suggested the role of complex III in nicotine-mediated inhibitory effect on the mitochondrial respiratory machinery [172]. Diaz and Raval showed that 16–21 days of nicotine treatment inhibited cortical mitochondrial complex IV enzyme activity in a recent study [164]. These results indicate that the effects on individual brain regions should be taken into consideration while studying the effects of nicotine and smoking on whole brain mitochondrial energy metabolism. 

Electrons that escape from the mitochondrial ETC can react with oxygen, resulting in the formation of ROS. SOD is a key enzyme responsible for the detoxification of these ROS. Increased oxidative stress in neurons involve several mechanisms including production of hydrogen peroxide and hydroxyl radicals [173], reduced activity of GPx and SOD [174], and increased level of malondialdehyde (MDA) [175]. Nicotine has been linked with the formation of excess ROS from that mitochondrial respiratory chain which can cause oxidative stress in the brain. Although a few studies have demonstrated that nicotine can decrease ROS formation [168], several other studies have shown that nicotine exposure may increase cytosolic ROS, resulting in enhanced oxidative stress [176,177,178]. Ande et al. showed similar results in astrocytes [179]. Seven days of nicotine treatment was shown to enhance ROS formation in the temporal cortex of the rat brain [176]. However, another study showed that nicotine increased ROS production in rat brain hippocampal region other than cortex [177] which was associated with DNA damage and lipid peroxidation. Das et al. also showed that in vivo nicotine treatment enhanced lipid peroxidation in different rat brain regions [178]. Furthermore, researchers have shown that nicotine exposure decreased the levels and functions of antioxidant enzymes such as SOD and glutathione-S-transferase [176,178], which would exacerbate brain oxidative stress. Additionally, both in vitro and in vivo studies have demonstrated ROS generation by electronic cigarette emission [180,181].

Particulate and gaseous elements in cigarette smoke other than nicotine are also responsible for vascular reactivity and endothelial dysfunctions. Kourembanas et al. showed that carbon monoxide could inhibit the production of endothelin-1 and platelet-derived growth factor B by endothelial cells [182]. Free radicals play a vital role in mediating smoking-induced endothelial injury [183]. Lipid peroxidation is the prominent mechanism of endothelial damage by free radicals [184]. Chronic nicotine-free cigarette smoke extract exposure causes endothelial dysfunctions by increasing vascular wall-derived superoxide generation [185]. Naik et al. used nicotine-free and low-nicotine cigarettes to show that these products caused significant BBB endothelial dysfunctions by enhancing the release of reactive oxygen and nitrogen species, downregulating the expression of tight junction proteins, and enhancing inflammatory response [186]. They also concluded that these observed toxic effects at the BBB endothelium correlated with the tar and nitric oxide levels in cigarettes. ROS from cigarette smoke are also involved with platelet activation [187]. Acrolein is a key component in tobacco smoke that can cause platelet aggregation, formation of platelet–leukocyte aggregates, and release of prothrombotic mediators from platelet granules thereby, enhancing the risk of thrombotic events [188]. Unsaturated aldehyde components of cigarette smoke, including acrolein and crotonaldehyde, have been shown to inhibit in vitro chemotaxis of human PMNs [189]. Further, aqueous extract of tobacco smoke decreased glycolytic function and phagocytic ability of PMNs, reducing their antibacterial function and increasing toxic effects [190].

The abovementioned smoking/nicotine-induced alterations in mitochondrial energy metabolism may add to the disrupted brain energy metabolism in ischemic stroke. Studies are needed to address this unexplored hypothesis. Mitochondria are also the essential source of ROS generation in cells which contribute to the pathogenesis of ischemia, reperfusion, and I/R injury [191]. A recent study showed that nicotine may cause brain injury by enhancing neuroinflammation after cerebral ischemia due to increased mitochondrial oxidative stress [28]. Another study suggested that chronic nicotine exposure could exacerbate transient focal cerebral ischemia-induced brain injury due to preexisting oxidative stress via increased superoxide level and reduced manganese superoxide dismutase (MnSOD) and uncoupling protein-2 levels in the cerebral cortex and arteries in rats [192]. Conversely, nicotine showed protective effects on brain mitochondrial respiration in an anoxia/reoxygenation model [193]. However, when nicotine was administered post anoxia but before reoxygenation, it was unable to preserve mitochondrial function [168]. The possible role of smoking and nicotine in brain mitochondrial energy metabolism in ischemic stroke is depicted in Figure 4.

## 6. Effects of Diabetes on Brain Energy Metabolism

A brief introduction to the characteristics of two major types of diabetes and different animal models employed to mimic these diseases is given below to help clarifying their effects on brain energy metabolism. Type 1 diabetes is characterized by a lack of insulin production, which is caused by autoimmune destruction of the pancreatic beta cells. Chemical induction (streptozotocin (STZ), alloxan), spontaneous autoimmune (NOD mice, BB rats), genetic induction (AKITA mice), and virally-induced (Coxsackie B virus, encephalomyocarditis virus) are different rodent models of type 1 diabetes [194]. 

In type 2 diabetes, there are insulin resistance and dysfunctional beta cells. In vivo models of type 2 diabetes include obese models (Lep ^ob/ob^ mice, Lepr ^db/db^ mice, KK mice), induced obesity (high fed diet, desert gerbil), non-obese models (GK rat), and genetically induced models (hIAPP mice, AKITA mice) [194]. The high fat diet (HFT) model of type 2 diabetes is characterized by obesity, insulin resistance, and altered glucose homeostasis. In recent times, type 2 diabetes models have been developed in rodents using STZ treatment in conjunction with HFT to better mimic human type 2 diabetes [195,196,197]. Use of multiple low dose injections of STZ after HFT pretreatment produces more stable type 2 diabetes models, which is characterized by hyperglycemia and insulin resistance. 

### 6.1. Glucose Utilization

Different studies have shown that experimental hyperglycemia and type 1 diabetes could decrease glucose transport across the BBB [198,199] which was connected with the intensity of the hyperglycemia. GLUT1 at the BBB was also shown to be downregulated with hyperglycemia [31]. Other studies have also reported decreased brain glucose utilization in diabetic rodents [32,200]. Glycolytic function was decreased in a type 1 diabetes model [32]. In contrast, another study showed that glycolytic activity was at least initially increased in a diabetogenic drug, STZ, -induced type 1 diabetes model [201]. Further, Li et al. reported that hyperglycemia could increase glycolytic metabolism, ATP, and glycogen content in primary astrocytes [202]. Brain glucose utilization was also shown to be decreased in type 2 diabetes [203,204], with reports of both downregulated [204] and unchanged GLUT1 expression [203] at the BBB. Neuronal GLUT3 expression was non-significantly decreased in an in vivo study with db/db diabetic mice [203]. With HFT, lower brain glucose uptake was observed in mice brain PET scans [205]. Liu et al. showed reduction of neuronal GLUT3 and GLUT4 in mice with 3 months of HFT treatment [206]. Moreover, decreased activity of the TCA cycle has been observed in experimental studies modeling type 2 diabetes [207], along with reduction in energy sources which may upregulate glycolysis and consequently lead to neuronal damage [30]. 

However, Garcia-Espinosa et al. reported that TCA cycle function was unchanged in type 1 diabetes [32], whereas Sickmann et al. demonstrated impaired TCA cycle activity in type 2 diabetes [208]. One of the subunits of the PDH enzyme complex was shown to be decreased in a type 2 diabetes model [209]. Additionally, another study reported that hyperglycemia, induced by intracerebroventricular application of STZ, decreased glycolytic enzymes and the alpha-ketoglutarate dehydrogenase enzymes [210]. Although preclinical models of type 1 and type 2 diabetes have limitations, the observed effect of diabetes on brain glucose utilization suggests that the coexistence of diabetes could adversely affect brain glucose utilization in ischemic stroke, thereby, worsening ischemic brain injury. These effects are illustrated in Figure 3. 

### 6.2. Mitochondrial Function and Oxidative Stress

It has been reported that diabetes is related to mitochondrial dysfunctions [211]. Diabatic encephalopathy causes mitochondrial alterations including increased level of ROS [212], lipid peroxidation, and nitrite, and reduced total antioxidant level [213]. In addition, diabetes-mediated oxidative stress is responsible for increasing proinflammatory cytokines which, in turn, leads to neuronal degeneration [214]. One study reported that diabetes impaired mitochondrial respiration in brain which could be due to alterations in ETC function and oxidative phosphorylation [215]. The inner membrane of mitochondria is the main target of ROS due to an elevated level of polyunsaturated fatty acid which leads to oxidation. Thus, lipid peroxidation and oxidative alterations in ETC result in variations in electron flow as well as electron leakage which consequently lead to generation of ROS and mitochondrial ETC damage [216]. Diabetes-induced ROS generation involves both enzymatic and non-enzymatic pathways. The components of enzymatic pathways include NOX, NOS, COX, LOX, cytochrome P450, XO, and myeloperoxidase (MPO). On th contrary, the non-enzymatic pathways include deficiencies in mitochondrial ETC, transition-metal catalyzed Fenton reactions, advanced glycation end (AGE) products, glucose autooxidation, and polyol (sorbitol) pathway [217,218]. Among the abovementioned sources, NOX is one of the primary sources of ROS generation in diabetic conditions involving different organs [218]. It has been demonstrated that hyperglycemia inhibited ETC at the levels of complexes III, IV, and V in STZ-induced type 1 diabetes rat model [219]. Moreover, the level of ROS, NO, and expression of mitochondrial NO synthase were found to be augmented in mitochondria, while activity of GSH peroxidase enzyme and protein content of MnSOD were found to be decreased. A decreased level of GSH and increased amount of GSSG were also found in this study [219]. Decreased activity of complexes III, IV, and V of the respiratory chain by oxidative and nitrosative stress and reduced ATP level can cause mitochondrial dysfunctions [219]. Hyperglycemia also activates the neurotoxic polyol pathway [220], resulting in excess ROS, reactive nitrogen species, and AGE products, and impaired Na^+^/K^+^-ATPase activity [221]. Further, hyperglycemia can excessively stimulate microglia and microangiopathy [221]. Hyperglycemia can also cause increased levels of mitochondrial NO and aconitase activity and a decreased level of mitochondrial lipid peroxidation at early stage of diabetes [222]. Thus, diabetes is related to an increased level of oxidative stress which affects lipid, membrane protein, and mitochondrial DNA which ultimately result in mitochondrial dysfunction [222]. Brain mitochondrial studies in a type 1 diabetes model also suggested decreased respiration with dysfunctional electron transport or ATP synthase [223]. Expression of antioxidant enzymes such as SOD, catalase, and glutathione peroxidase are also decreased in the diabetic brain [224]. 

Mitochondria play a crucial role in maintaining ischemic brain physiology in diabetes [225]. As mitochondria are necessary for ATP generation, calcium influx buffering, free radical production, and proapoptotic factors release in ischemic brain [226,227], mitochondrial dysfunction in ischemic brain may lead to increased level of intracellular calcium, oxidative stress generation, and reactive peroxynitrite species production [225]. Moreover, concomitant mitochondrial permeability transition pore activation facilitates the release of cytochrome c that stimulates the apoptotic cell death pathways [228]. Consequently, this may results in the stimulation of terminal executioner caspases and apoptotic cell death, which is considered to be one of the important mechanisms of cell death in ischemic stroke [229]. As impaired mitochondrial function has also been observed in diabetes, coexistence of this condition may exacerbate the alterations of mitochondrial energy metabolism and enhance oxidative stress in ischemic stroke. Karasu et al. reported that diabetes upregulated auto-oxidation of glucose which caused protein glycation non-enzymatically and this ultimately resulted in increased level of ROS which participate in ischemic brain injury [152]. Glucose auto-oxidation, stimulated polyol pathway, AGE synthesis, and endogenous antioxidant enzyme inhibition may play crucial roles in the enhancement of brain oxidative stress in diabetic models [230,231,232]. Additionally, disparity between free radical generation and their quenching by endogenous antioxidant enzymes has been found in diabetes, which may damage proteins, lipids, and nucleic acids in neurons [233]. In summary, it can be said that diabetes is responsible for enhanced electron flux to complex III, impaired antioxidant enzyme system, reduced ATP production, decreased calcium accumulation, and accelerated mitochondrial fission, which may result in increased oxidative stress, lower availability of ATP during ischemia-reperfusion, mitochondrial swelling, and proapoptotic molecules’ release. Collectively, these may worsen brain injury and neurological outcomes of ischemic stroke [225]. Figure 4 depicts these effects. Understanding the changes in ischemic brain energy metabolism with coexisting smoking and/or diabetes, coupled with novel drug delivery strategies [234] can facilitate development of specific therapeutic interventions for these conditions. 

## 7. Conclusions

Altered glucose utilization and impaired mitochondrial functions contribute to defective brain energy metabolism and enhanced oxidative stress in ischemic stroke. Comorbid conditions such as nicotine/smoking exposure and diabetes can further alter brain energy metabolism, thereby, worsening the outcome of an ischemic stroke event in a growing population of patients. Since there are, so far, few studies that hae addressed the outcome of stroke in appropriate pre/comorbid models, particularly addressing smoking via inhalation, a scientific gap exists which needs to be addressed. Until this is addressed, the stroke field will continue to struggle in correlating preclinical results to the human stroke condition.

## Figures and Tables

**Figure 1 ijms-23-08512-f001:**
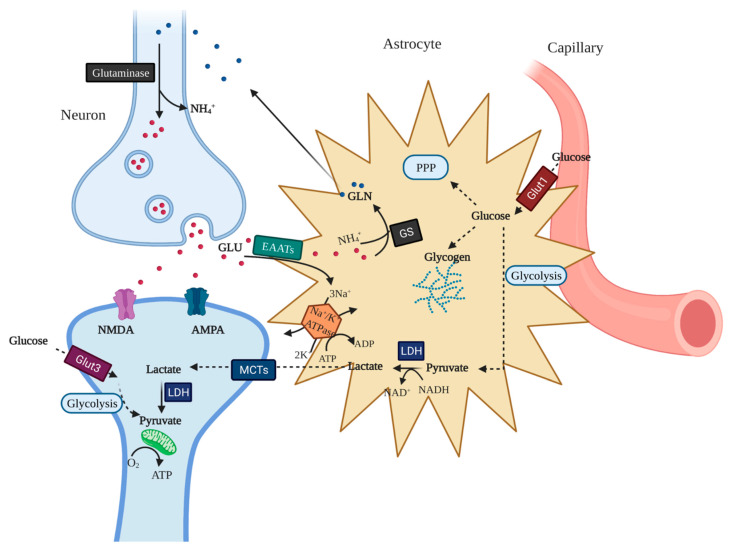
Interplay of glucose metabolism in neurons and astrocytes (created in BioRender).

**Figure 2 ijms-23-08512-f002:**
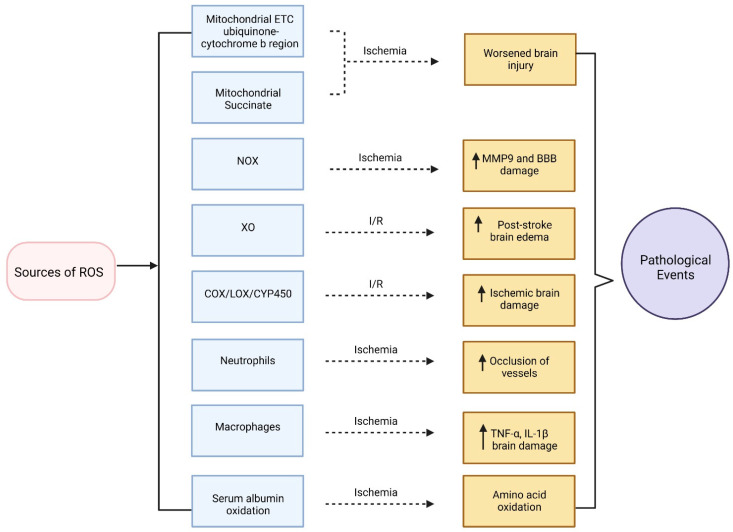
Sources of reactive oxygen species (ROS) and their perturbations in ischemic stroke.

**Figure 3 ijms-23-08512-f003:**
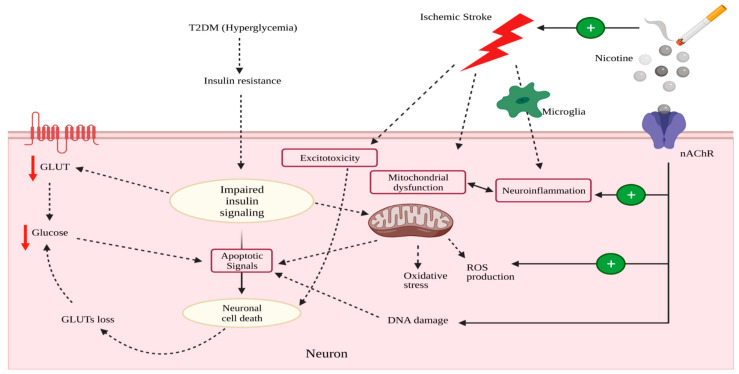
Effects of nicotine/smoking and diabetes on neuronal glucose utilization in ischemic stroke (created in BioRender).

**Figure 4 ijms-23-08512-f004:**
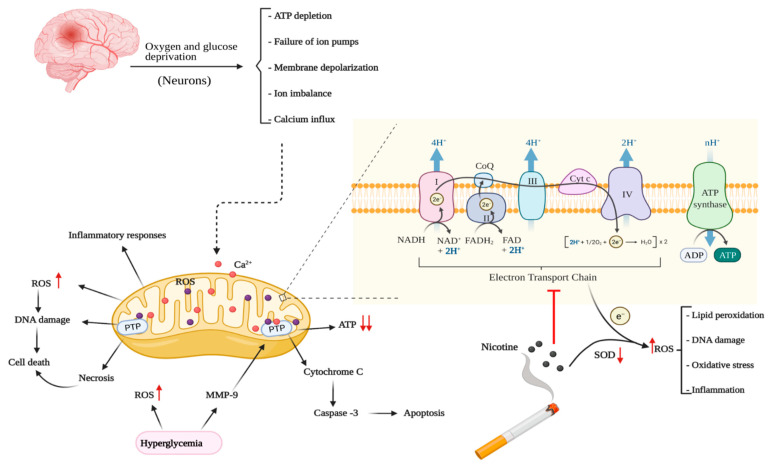
Effects of nicotine/smoking and diabetes on mitochondrial energy metabolism of neurons in ischemic stroke (created in BioRender).

**Table 1 ijms-23-08512-t001:** Targeting mitochondria for regenerative therapy in ischemic stroke.

Pharmacologic Class	Description/Mechanism of Action(s)	Stroke Model Used	Species and Number of Animals	Sex of Animals	Therapeutic Outcome(s)	Year Published with Reference(s)
Mitochondrial fission inhibitor (Mdivi)	-Inhibition of the assembly of Drp1 and GTPase Drp1 enzymatic activity -Reduction of the level of Drp1 and Cytochrome c	tMCAO	Wistar rats, 20/group	Male	-Reduced cerebral damage induced by ischemia-reperfusion injury -Inhibition of apoptotic cell deaths	2013 [84], 2014 [85]
Modulators of purinergic receptors	-Stimulation of glia-specific purinergic receptor, P2Y_1_R -Increased mitochondrial O_2_ consumption and ATP production -P2X7R antagonism decreased expressions of P2X7R, NLRP3, ASC, Caspase-1 p20, and cleaved caspase-3 in ischemic brain tissue	Photothrombotic model	- Transgenic mice -C57BL/6J mice, 3 or 6/group	Male	-Reduced neuronal damage, cell death, and swelling in ischemic stroke -Reduced brain infarct size and neuronal apoptosis -Improved functional outcome after stroke	2013 [86], 2017 [87]
Antioxidants and SOD mimetics	-Free radical trapping -Mitochondria-specific reduction of O_2_^−^, cytochorme c, caspase-3, and CHOP -Inhibition of the NF-κB pathway	-pMCAO -tMCAO	C57BL/6J mice; 6/group, Wistar rats, 12/group	Male	-Decresed brain lesion volume, motor impairment, and neglect in animal models -Reduced brain infarct volume, tissue damage, and apoptosis	2007 [88], 2009 [89], 2012 [90,91], 2022 [92]
Activators of NAD-dependent deacetylase sirtuin 1 (SIRT1)	-Reduction of inflammation and oxidative stress -Prevention of lipid peroxidation -Mimicking ischemic preconditioning in brain -Alteration of CDK5R1/SIRT1 signaling	Global cerebral ischemia followed by asphyxial cardiac arrest	Sprague Dawley (SD) rats, 5 or 8/group C57BL/6J mice, 15/group	Male	-Reduced brain infarct volume and neurological deficits -Improved regional brain blood flow, apoptosis, and mitochondrial dysfunctions	2009 [93], 2012 [94], 2022 [95]
Methylene blue	-Alternative electron carrier which reduces electron leakage and ROS production -Enhancing mitochondrial oxygen consumption rate and decreasing the extracellular acidification rate	-tMCAO -Global hypoxia (15% O_2_)	-Sprague-Dawley rats -Sprague-Dawley rats, 6/group	Male	-Reduced ischemic brain infarct volume	2011 [96], 2013 [97]
Melatonin	-Enhancing the expression of neuronal bcl-2 -Inhibition of autophagy -Activation of the PI3K/Akt pro-survival pathway -Reduction of oxidative stress -Inhibition of MAPK pathway	-tMCAO -BCO	-Rats -Mongolian gerbils, 10/group	Male	-Decreased brain infarct area and neurological impairments -Reduction of post-ischemic brain area -Increased survival and reduced hyperactivity	1999 [98], 2000 [99], 2014 [100], 2022 [101]
Hydrogen sulfide (H_2_S)	-Stimulation of ATP-sensitive potassium channel/protein kinase C/extracellular signal-regulated kinase/heat shock protein 90 pathway -Inhibition of ROS and caspase-3	Four artery occlusion	Sprague-Dawley rats, 6/group	Male	-Neuroprotection in ischemic neurons -Reduction of neuronal apoptosis	2010 [102], 2013 [103,104]
Alpha-phenyl-*tert*-butyl-nitrone (PBN)	-Free radical scavenger -Improved mitochondrial respiratory function	-Total cerebral ischemia -tMCAO	-Fischer 344 rats, 3-5/group -Sprague-Dawley rats, 6-7/group	Male	-Improved neurological performance	2008 [105,106], 2010 [107]
Luteolin	-Decrease in ROS production -Protecting the activities of mitochondria, catalase, and glutathione -TNF signaling pathway	-tMCAO -pMCAO	Sprague-Dawley rats, 16-18/group, Sprague-Dawley rats, 6-10/group	-Female -Male	-Reduced brain infarct volume -Improved behavioral and motor functions after stroke	2011 [108], 2012 [109,110], 2021 [111]
Selenium compounds	-Reduction of oxidative stress (ROS, malondialdehyde) and proinflammatory cytokines -Protection of mitochondrial dehydrogenase and complex I activity and reduced mitochondrial swelling -Decreased autophagy	-tBCCAO -tMCAO	Wistar rats, 8/group Diabetic Sprague-Dawley rats, 20/group	Male	-Improved brain infarct and edema -Decreased BBB damage -Improved neurological functions	2012 [112], 2014 [113], 2021 [114]

tMCAO, transient middle cerebral artery occlusion; pMCAO, permanent middle cerebral artery occlusion; SOD, superoxide dismutase; ROS, reactive oxygen species; BCO, bilateral carotid occlusion; tBCCAO, transient bilateral common carotid artery occlusion; BBB, blood-brain barrier

## Data Availability

Not applicable.

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
