# Peer review of "Brain Energy Metabolism in Ischemic Stroke: Effects of Smoking and Diabetes"

_ijms, 2022, doi:10.3390/ijms23158512_

Round 1
Reviewer 1 Report
The manuscript by Sifat and colleagues is a comprehensive review that outlines how brain energy metabolism is affected in the setting of ischemic stroke. The authors take a mechanistic standpoint and describe, in rigor detail, specific biochemical pathways that are responsible for energy production in the brain. Of particular note, the authors describes how stroke alters these pathways and how co-morbidities (i.e., smoking, diabetes mellitus) can further exacerbate impairments in brain energy metabolism. This is an excellent and timely review; however, some English language editing is required due to a few grammatical errors. Additionally, I have some specific comments on scientific content for the authors to consider.
1. It is well-known that microglial are metabolically flexible and can rapidly adapt to changes in their external environment. As noted by Bernier and colleagues (Nat Commun. 11: 1559, 2020), microglia can consume glutamine as a metabolic fuel in pathologies where glucose is absent. This enables microglia to continue to provide immune support to the brain under hypoglycemic or aglycemic conditions. The authors should include a discussion on the metabolic flexibility of microglia and how this could affect stroke pathogenesis.
2. Lines 352-403 - the authors provide detailed information on oxidative stress pathways and their association with cerebral ischemia. It would be helpful to include a figure depicting these pathways and how they are perturbed in the setting of stroke.
3. Section 5.1 - the authors describe several studies on effects of acute/chronic nicotine exposure and local cerebral glucose utilization. In this section, the authors should explicitly describe the nicotine doses used in these studies and how they relate to nicotine content in currently marketed tobacco products and/or nicotine exposure in smokers.
4. Lines 557-578 - several studies have described the use of streptozotocin in the establishment of type II diabetes animal models (Qian et al. PLoS One. 10(8): e0136390, 2015; Premilovac et al. Sci Rep. 7: 14158, 2017; Furman. Current Protocols. 1:e78, 2021). Such a chemical induction strategy is often done in conjunction with a high-fat diet. Therefore, the use of streptozotocin in the context of type 2 diabetes should be discussed by the authors.
Author Response
RESPONSE TO REVIEWER’S COMMENTS (Reviewer #1)
Comments and Suggestions for Authors
The manuscript by Sifat and colleagues is a comprehensive review that outlines how brain energy metabolism is affected in the setting of ischemic stroke. The authors take a mechanistic standpoint and describe, in rigor detail, specific biochemical pathways that are responsible for energy production in the brain. Of particular note, the authors describes how stroke alters these pathways and how co-morbidities (i.e., smoking, diabetes mellitus) can further exacerbate impairments in brain energy metabolism. This is an excellent and timely review; however, some English language editing is required due to a few grammatical errors.
Response: Thank you very much for agreeing to review this manuscript. The manuscript is thoroughly edited now for English language and grammatical error correction.
Additionally, I have some specific comments on scientific content for the authors to consider.
- It is well-known that microglial are metabolically flexible and can rapidly adapt to changes in their external environment. As noted by Bernier and colleagues (Nat Commun. 11: 1559, 2020), microglia can consume glutamine as a metabolic fuel in pathologies where glucose is absent. This enables microglia to continue to provide immune support to the brain under hypoglycemic or aglycemic conditions. The authors should include a discussion on the metabolic flexibility of microglia and how this could affect stroke pathogenesis.
Response: Thank you for your insightful comment. We have revised the manuscript to include a discussion on the metabolic flexibility of microglia and its potential role in stroke pathogenesis in page 11 (lines 426-439).
- Lines 352-403 - the authors provide detailed information on oxidative stress pathways and their association with cerebral ischemia. It would be helpful to include a figure depicting these pathways and how they are perturbed in the setting of stroke.
Response: A figure has been included in the revised manuscript according to your suggestion (page 11).
- Section 5.1 - the authors describe several studies on effects of acute/chronic nicotine exposure and local cerebral glucose utilization. In this section, the authors should explicitly describe the nicotine doses used in these studies and how they relate to nicotine content in currently marketed tobacco products and/or nicotine exposure in smokers.
Response: Nicotine doses have been provided for those studies with their correlation with nicotine level in smokers in section 5.1, as suggested (page 12).
- Lines 557-578 - several studies have described the use of streptozotocin in the establishment of type II diabetes animal models (Qian et al. PLoS One. 10(8): e0136390, 2015; Premilovac et al. Sci Rep. 7: 14158, 2017; Furman. Current Protocols. 1:e78, 2021). Such a chemical induction strategy is often done in conjunction with a high-fat diet. Therefore, the use of streptozotocin in the context of type 2 diabetes should be discussed by the authors.
Response: Thank you for your valuable comment. The use of streptozotocin in the context of type 2 diabetes has been discussed in the revised manuscript (page 15, lines 617-620).
Reviewer 2 Report
The manuscript entitled "Brain Energy Metabolism in Ischemic Stroke: Effects of Smoking & Diabetes” is an interesting review focusing on the study of brain energy metabolism related to ischemic stroke and comorbid with smoking and diabetic conditions.
· The manuscript needs some copy-editing in certain parts.
· Some sentences get repeated in the introduction (for e.g., line 114-115). These can be rephrased in a different manner.
Author Response
RESPONSE TO REVIEWER’S COMMENTS (Reviewer #2)
The manuscript entitled "Brain Energy Metabolism in Ischemic Stroke: Effects of Smoking & Diabetes” is an interesting review focusing on the study of brain energy metabolism related to ischemic stroke and comorbid with smoking and diabetic conditions.
Response: Thank you very much for agreeing to review our manuscript.
- The manuscript needs some copy-editing in certain parts.
Response: The manuscript has been thoroughly edited for English language and grammatical errors in the revised manuscript.
- Some sentences get repeated in the introduction (for e.g., line 114-115). These can be rephrased in a different manner.
Response: Thank you for your comment. The manuscript has been revised to a more concise format by omitting repeated or extraneous information. As per your suggestion, the abovementioned sentence is rephrased in the revised manuscript (line 87-89) as “Oxidative stress and mitochondrial dysfunctions are key factors contributing to hyperglycemia-induced enhanced ischemic brain injury.”
Reviewer 3 Report
The Authors in this manuscript described two of the most common established and modifiable cerebrovascular risk factors. Ischemic stroke is indeed more prevalent in the diabetic and smoker populations.
The additive effects of these two risk factors are not limited to the metabolic rewiring of the system.
The review has some major issues to be assessed:
1. The paper starts with a long and generic introduction composed of the first three paragraphs in which no specific or innovative mechanisms are elucidated but just simplified biochemistry and cellular biology.
2. The regenerative therapies described in ischemic stroke consider a wide range of molecules with a putative unspecific or generic mechanisms of action and "therapeutic" outcomes that are misleading.
3. Models of diabetes are enlisted byt without a clear purpose for this review
4. The final mechanism that is proposed should be the true topic of the review, to be fully detailed without all the aforementioned unnecessary paragraphs. However considering all the other possible interference of these risk factors, is highly difficult to insolate the pure ROS-mediated damage involving the mitochondrial respiratory chain.
Minor concern: citations should be checked, see line 420
Author Response
RESPONSE TO REVIEWER’S COMMENTS (Reviewer #3)
The Authors in this manuscript described two of the most common established and modifiable cerebrovascular risk factors. Ischemic stroke is indeed more prevalent in the diabetic and smoker populations.
The additive effects of these two risk factors are not limited to the metabolic rewiring of the system.
Response: Thank you very much for agreeing to review our manuscript.
The review has some major issues to be assessed:
- The paper starts with a long and generic introduction composed of the first three paragraphs in which no specific or innovative mechanisms are elucidated but just simplified biochemistry and cellular biology.
Response: Thank you for your comment. The manuscript has been revised to make the introductory paragraphs more concise by replacing simplified biochemistry and molecular biology with specific mechanisms (page 1-2).
- The regenerative therapies described in ischemic stroke consider a wide range of molecules with a putative unspecific or generic mechanisms of action and "therapeutic" outcomes that are misleading.
Response: The table 1 has been substantially modified in the revised manuscript (page 7-9) to include specific and detailed mechanisms of actions and therapeutic outcomes of the discussed molecules.
- Models of diabetes are enlisted but without a clear purpose for this review.
Response: Thank you for your comment. We think it would be helpful for the readers to know about the characteristics of the 2 major types of diabetes and the different animal models used to mimic those conditions to understand the effects of diabetes on brain energy metabolism, which is explained in the revised manuscript (page 15, line 605-607). However, the discussion about the models of diabetes is modified to a more concise format in the revised manuscript (page 15).
- The final mechanism that is proposed should be the true topic of the review, to be fully detailed without all the aforementioned unnecessary paragraphs. However considering all the other possible interference of these risk factors, is highly difficult to insolate the pure ROS-mediated damage involving the mitochondrial respiratory chain.
Response: We agree with your comment and have revised the entire manuscript to a more concise format and focused more into ROS-mediated damage (e.g., a new figure has been included) to reflect your concern as much as possible.
Minor concern
- Citations should be checked, see line 420.
Response: Thank you very much for noticing this error. The citations are checked thoroughly and the error in line 420 (line 471 in the revised manuscript), along with others (lines 309, 514) are corrected.
Round 2
Reviewer 1 Report
The authors have addressed all comments from the previous critique. The revised manuscript is much improved and represents a valuable contribution to the field.
Author Response
RESPONSE TO REVIEWER’S COMMENTS (Reviewer # 1)
- The authors have addressed all comments from the previous critique. The revised manuscript is much improved and represents a valuable contribution to the field.
Response: Thank you very much for your valuable review of our manuscript which has helped us to improve it.